# THE ERA OF REAL-WORLD HUMAN INTERACTION: RL FROM USER CONVERSATIONS

## ABSTRACT

We posit that to achieve continual model improvement and multifaceted alignment, future models must learn from natural human interaction. Current conversational models are aligned using pre-annotated, expert-generated human feedback. In this work, we introduce Reinforcement Learning from Human Interaction (RLHI), a post-training paradigm that learns directly from in-the-wild user conversations. We develop two complementary methods: (1) *RLHI with User-Guided Rewrites*, which revises unsatisfactory model outputs based on users' natural-language follow-up responses, (2) *RLHI with User-Based Rewards*, which learns via a reward model conditioned on knowledge of the user's long-term interaction history (termed persona). Together, these methods link long-term user personas to turn-level preferences via persona-conditioned preference optimization. Trained on conversations derived from WildChat, both RLHI variants outperform strong baselines in personalization and instruction-following, and similar feedback enhances performance on reasoning benchmarks. These results suggest organic human interaction offers scalable, effective supervision for personalized alignment.

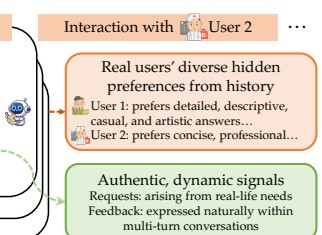

**Learning from Annotated Human Feedback**

Static, isolated queries

**Prompt**: How can I develop a habit of drawing daily?

**Response 1 (chosen):** Developing a daily habit of drawing can be challenging but…
**Response 2 (rejected):** As an AI language model, I cannot develop habits…
**Meta data**: scores, ranking, reasons, etc.

Preference labels from annotators

**Learning from Organic Human Interaction**

Interaction with User 1 | Interaction with User 2 ⋯

I'm sketching a city at night. what details should I highlight?

Show glowing windows, neon signs, rivers of car headlights…

Can you give more details?

Real users' diverse hidden preferences from history
User 1: prefers detailed, descriptive, casual, and artistic answers…
User 2: prefers concise, professional…

Authentic, dynamic signals
Requests: arising from real-life needs
Feedback: expressed naturally within multi-turn conversations

Figure 1: **From annotated feedback to the *era of real-world human interaction*. Left:** Traditional alignment relies on expert-curated annotations of ranked responses or labels, providing static, out-of-distribution supervision. **Right:** In-the-wild conversations reveal users' long-term histories, dynamic demands, and diverse signals, enabling personalized, contextual, and continual learning.

## 1 INTRODUCTION

Today, language model post-training primarily depends on static corpora of expert-annotated data: verifiable questions, fixed demonstrations, and rankings or ratings collected outside of natural conversational contexts. While these datasets are effective for instilling general capabilities, they reflect the opinions and heuristics of annotators in unnatural scenarios rather than the *authentic, diverse long-term goals and preferences of real users*; they capture static, context-free judgments instead of *evolving, situational demands*; and they scale with labeling budgets rather than with actual usage and diversity of organic users, as is illustrated on the left side of Figure 1.

In contrast, humans learn and improve through continual experience by interacting with their environment and other actors, receiving feedback, and adjusting behavior over time (Tomasello et al., 2005). Likewise, a rich and organic source of supervision for language models already exists in the wild: **human interaction**—the ongoing, natural exchanges between models and real users. As is shown on the right side of Figure 1, such organic interactions reveal hidden user preferences from

long-term histories and dynamic, context-dependent demands, as people reveal their priorities and concerns not through annotation formats, but by discussing what matters to them, revising or re-attempting questions, explicitly or implicitly approving or critiquing model outputs, following up, or switching goals mid-dialogue. Because they arise directly from model outputs in authentic usage contexts, such interactions provide a rich signal for learning personalized and adaptable behavior, paving the way toward personal superintelligence. While this source of supervision has historically been hard to extract, resulting in resorting to collecting static training data instead, the power of modern language models now gives us a greater ability to extract these signals.

To achieve this vision, we introduce RLHI, a post-training paradigm that learns directly from in-the-wild conversations through two complementary methods: (1) *RLHI with User-Guided Rewrites* (§2.3), which revises unsatisfactory model outputs based on users' natural-language follow-ups, and pairs the rewrites with the originals for preference learning; and (2) *RLHI with User-Based Rewards* (§2.4), which ranks candidate responses using a reward model conditioned on user personas derived from long-term histories to generate preference pairs. Together, these methods link long-term user personas to turn-level preferences via persona-conditioned preference optimization.

We evaluate RLHI in three settings. (i) *User-based evaluation* with our WILDCHAT USEREVAL: both RLHI variants outperform strong baselines in personalization and instruction-following, and a human study corroborates these trends. (ii) *Standard instruction-following benchmarks*: *User-Based Rewards* attains a 77.9% length-controlled win rate on AlpacaEval 2.0, surpassing all RLHF methods. (iii) *Reasoning*: *User-Guided Rewrites* raises average accuracy from 26.5 to 31.8 across four benchmarks. Our ablation studies further show that RLHI benefits from user guidance and interaction diversity, that reinforcement learning outperforms supervised finetuning, and that quality filtering is essential for effectively leveraging noisy human interaction data.

## 2   RLHI: REINFORCEMENT LEARNING FROM HUMAN INTERACTION

### 2.1   THE ERA OF REAL-WORLD HUMAN INTERACTION

Artificial intelligence (AI) has progressed rapidly in recent years through large-scale pretraining and fine-tuning with human examples and preferences. Yet this trajectory is slowing: high-quality data is running out, and imitation alone cannot push systems beyond existing human knowledge. Recent proposals call for an *era of experience* (Silver & Sutton, 2025), in which AI systems advance by continually learning from their own interactions with the world. Since these systems ultimately exist to assist humans, interaction with users becomes a natural and essential dimension of this shift. The *era of real-world human interaction* thus forms a core pillar of the era of experience, providing both the raw data and personalization signals necessary for adaptive, human-centered intelligence.

We define learning from human interaction as the process of improving AI models through natural, continual exchanges with real users. Such interactions may involve messages, actions, requests, or demonstrations provided in direct response to the model's outputs. These exchanges not only reveal user goals and preferences but also create an evolving feedback loop that enables systems to refine their behavior over time. To truly benefit from human interaction, AI needs to go beyond coarse binary labels to absorb knowledge, preferences, reasoning skills, perceptual cues, cooperative strategies, and social norms, learning deeper forms of intelligence through interaction.

Compared with other training data sources, human interaction is distinguished by three key properties: (1) **Contextual grounding** — arises within the flow of ongoing tasks or conversations, directly tied to the user's situational needs and the model's prior outputs, while being shaped by personalized knowledge of the user's profile, history, and preferences; (2) **Evolving distribution** — reflects goals that shift, environments that change, and preferences that adapt over time, thereby providing supervision that is temporally relevant and aligned with the real distribution of human needs and priorities; and (3) **Diverse supervision signals** — appears in both explicit high-bandwidth signals beyond scalar rewards (e.g., corrections or clarifications) and implicit cues (e.g., disengagement or frustration), and may include style and role assignments, emotional tone, or even adversarial inputs such as jailbreak attempts, which require careful handling, but also offer valuable information.

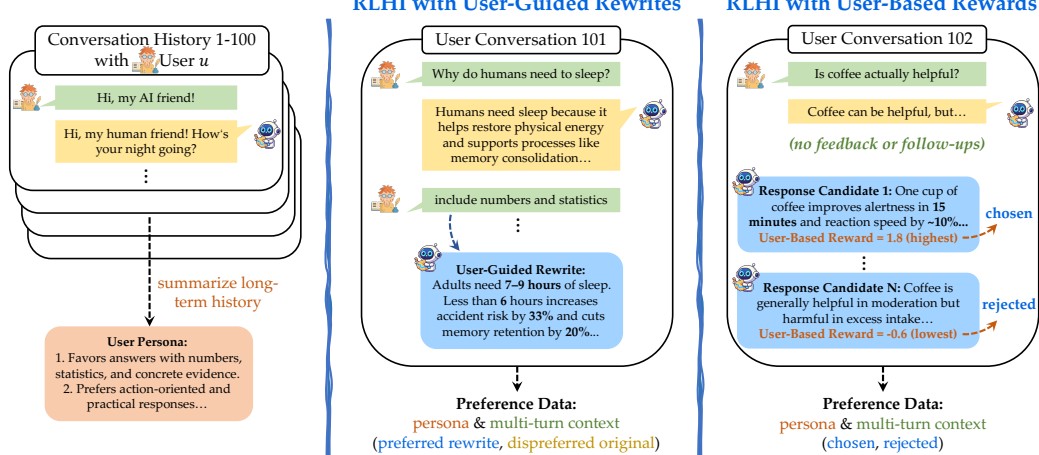

Figure 2: **Reinforcement Learning from Human Interaction (RLHI).** We derive a natural-language persona summary from each user's long-term conversational history. For real-world requests, RLHI operates in two modes: (1) User-Guided Rewrites, where unsatisfactory model outputs are revised based on users' natural-language follow-ups, creating preference pairs between the original and rewritten responses; and (2) User-Based Rewards, where multiple candidate responses are generated and ranked by a reward model conditioned on the user's persona, yielding chosen-rejected pairs. Both methods leverage personas and multi-turn context to enable personalized alignment.

In this paper, we focus on large language models that engage daily with millions of users. Here, human interaction takes the minimal form of textual messages, yet still conveys contextual, dynamic, and diverse requests, holding unique potential as a driver of continual model improvement.

## 2.2 ANALYSIS OF REAL-WORLD HUMAN INTERACTION

To determine the feasibility of our approach, we first consider *currently available human interaction data*, analyzing its properties. We note that these properties are necessarily tied to the capabilities of current models, and we expect these statistics to change considerably in the coming years.

**Users often provide feedback to improve model responses.** We analyze user messages in the WildChat-1M dataset, which contains over one million conversations with ChatGPT (Zhao et al., 2024b). In each multi-turn conversation, the first message is the *initial request*, and we prompt an GPT-4o model to classify user follow-up messages into four types: (1) *new requests*, where the user shifts to a new topic, substantially reformulates the original, or provides unrelated input; (2) *re-attempts with feedback*, where the user refines the initial prompt, adds clarification, or provides explicit or implicit feedback; (3) *re-attempts without feedback*, where the same prompt is repeated with no new input; and (4) *positive feedback*, where the user expresses praise or satisfaction. We find the distributions are: 27.07% of user messages are *initial requests*, 40.40% are *new requests*, 26.51% are *re-attempts with feedback*, 4.77% are *re-attempts without feedback*, and 1.25% are *positive feedback*, with more details and examples in Appendix A. Conversations of later stages are dominated by *re-attempts with feedback*, accounting for 83.15% of user utterances after the fifth turn. *re-attempts with feedback* are relatively short, averaging 272 characters compared to 725 for initial requests, but are semantically dense. We note that given the huge amount of human interactions in current production systems, these percentages convert to very large amounts of supervisory data. We note that while these are current statistics, in the future, as models display further capabilities, users will change their behavior. For example, if users know that models will learn from their textual feedback, then they are even more likely to provide it.

**Real-world human interaction data are more diverse than existing preference datasets.** Conversation messages span a wide range of forms and topics (for example, creative writing, analysis, and coding) and occur in conversations of highly varying length (average 2.54 turns). To quantify this diversity, we compare request contexts in our generated preference dataset with two widely used annotated feedback datasets: HH-RLHF (Bai et al., 2022) and HelpSteer2 (Wang et al., 2024b). From each dataset, we sample 500 examples, embed their contexts using OpenAI's text-embedding-

3-small model (OpenAI, 2024), and compute average pairwise cosine distances. WildChat users show the greatest contextual diversity (0.865), compared to 0.751 for HH-RLHF and 0.848 for Help-Steer2. These results suggest that real user interactions not only reflect authentic everyday needs but also span broader contexts and requests. Additional visualizations are provided in Appendix B.

**User personas are diverse with distinct characteristics.** We restructure the dataset by user and construct natural-language *personas* that summarize each individual's preferences from their conversation histories (see prompt in Figure 7). We observe that: (1) some users provide little feedback, while others reveal clear and consistent behaviors; (2) many personas reflect common expectations, yet a notable subset exhibit unique preferences (e.g., repeatedly requesting analogies or engaging in role-play with recurring characters); and (3) some users needs vary across domains (e.g., preferring step-by-step reasoning in math but quick takeaways in daily advice) or show evolving needs over time. To study these patterns, we examine several of the most frequently mentioned preference dimensions: expertise, desired informativeness, tone, and response structure. As shown in Table 1, majorities tend to prefer expert, expansive, serious, and well-structured responses, yet substantial portions favor the opposite qualities, underscoring the need to model both dominant trends and less common preferences.

Table 1: User preferences across conversational dimensions, based on a random subset of 5,000 WildChat users. Percentages represent the proportion of users with a clear preference. "Pct. None" denotes the percentage of users with no clear preference.

| Dimension | Preference 1 | Pct. | Preference 2 | Pct. | Pct. None |
|---|---|---|---|---|---|
| expertise | responses that can be easily understood by beginners | 24.1% | responses with expert-level knowledge | **59.8%** | 16.1% |
| informativeness | concise responses, without being verbose | 36.0% | expansive and informative responses, without missing background information | **49.9%** | 14.1% |
| tone | casual, friendly, and humorous responses | 4.9% | serious, formal, and professional responses | **84.5%** | 10.6% |
| structure | structured responses, with a clear and logical flow | **77.1%** | free-form responses, with a casual and conversational style | 9.1% | 13.8% |

## 2.3 RLHI with User-Guided Rewrites

In real-world scenarios, conversational models can generate unsatisfactory outputs—responses that are unhelpful, off-target, or misaligned with user intent. Organically, in such interactions, users frequently react by providing follow-up requests or explicit/implicit feedback (e.g., "Could you provide more details?"), signaling both dissatisfaction and expectations for improvement. Rather than reducing such feedback into coarse binary labels, we seek to exploit its rich semantic content. Leveraging feedback to help the model identify where it falls short and apply targeted updates provides a natural path toward more useful and better-aligned model behavior.

We rely on our user message classification in Section 2.2 to identify *re-attempts with feedback*, which make up 26.51% of all user messages in WildChat. In these cases, the model is prompted to revise its previous unsatisfactory response using the explicit or implicit user feedback (e.g., as in Figure 2, adding numbers and statistics when requested). The prompt we use is provided in Appendix Figure 8. This produces preference pairs where the user-guided rewrite is favored over the original output, directly reflecting user-indicated improvements.

To better ground learning in long-term user preferences, we prompt the LLM to summarize each user's latent preferences from their conversation histories into a user persona. These personas are incorporated into preference pairs generated via user-guided rewrites during training, and dynamically updated at inference time to guide personalized generation, as shown in Figure 9. The persona distills long-context signals into a compact representation, while turn-level feedback offers immediate, response-specific supervision. Together, long-context persona modeling and local feedback signals help the system capture user-specific expectations and styles that may differ from general preferences, linking a user's enduring preferences to desirable outputs.

To ensure the quality of preference pairs, we filter the data using two criteria: (1) User-guided rewrites must improve upon the original. We discard any rewrites with a user-based reward (details in Section 2.4) lower than the original to avoid harmful follow-ups. (2) Overall quality must be high. We apply the filtering techniques from RIP (Yu et al., 2025), with details provided in Appendix C.2.

Formally, for each training instance $i$ from user $u$, we consider the persona $p_u$, the multi-turn context $x_{u,i}$, a dispreferred original $y_{u,i}^-$, and a preferred rewrite $y_{u,i}^+$. We perform preference optimization using persona-conditioned Direct Preference Optimization (DPO), which maximizes the relative preference for $y_{u,i}^+$ over $y_{u,i}^-$ conditioned on both the prompt and persona:

$$\mathcal{L}_{\text{persona-DPO}} = \mathbb{E}_{u,i} \left[ \log \sigma \Big( \beta \Big( \log \frac{\pi_\theta(y_{u,i}^+|x_{u,i},p_u)}{\pi_{\text{ref}}(y_{u,i}^+|x_{u,i},p_u)} - \log \frac{\pi_\theta(y_{u,i}^-|x_{u,i},p_u)}{\pi_{\text{ref}}(y_{u,i}^-|x_{u,i},p_u)} \Big) \Big) \right], \tag{1}$$

where $\pi_\theta$ is the current policy, $\pi_{\text{ref}}$ a frozen reference model (a copy of the base model used as a baseline), and $\beta$ controls the sharpness of preference learning. This objective explicitly conditions preference optimization on user personas, aligning generation with individualized expectations derived from long-term interactions, and yielding more personalized, satisfactory responses.

## 2.4 RLHI with User-Based Rewards

In real-world human-LLM interactions, many initial requests do not come with follow-ups or feedback clarifying expectations for improvement. Nevertheless, these requests still reflect genuine user needs and are grounded in authentic human personas. Our goal is to improve model responses for such cases in a personalized manner. Using a (user-based) reward model provides a scalable way to learn from one-shot requests, enabling adaptation even when explicit feedback is absent.

To this end, we develop user-based rewards to guide model learning. For each user request, we generate preference pairs by first sampling $N$ candidate responses, then evaluating them with a reward model that explicitly conditions on the corresponding user persona. For example, as illustrated in Figure 2 (right), if long-term interactions indicate that a user favors answers with numbers, statistics, and concrete evidence, the reward model will assign higher scores to responses that not only meet general quality criteria but also reflect these user-specific characteristics.

Formally, for each training instance $i$ from user $u$, let $p_u$ denote the user persona and $x_{u,i}$ the multi-turn context. The LLM $\mathcal{M}$ generates $N$ candidate responses conditioned on both context and persona. A reward model $r$ then scores each candidate given $(x_{u,i}, p_u)$. Preference pairs $(y_{u,i}^+, y_{u,i}^-)$ are formed by selecting the highest- and lowest-scoring candidates:

$$\{y_{u,i}^{(n)}\}_{n=1}^N \sim \mathcal{M}(x_{u,i}, p_u) \quad \text{then} \quad \begin{cases} y_{u,i}^+ &= \arg\max_{n\in[N]} \ r\Big(y_{u,i}^{(n)} \mid x_{u,i}, p_u\Big), \\ y_{u,i}^- &= \arg\min_{n\in[N]} \ r\Big(y_{u,i}^{(n)} \mid x_{u,i}, p_u\Big). \end{cases} \tag{2}$$

We then apply persona-conditioned preference optimization, maximizing the relative preference for $y_{u,i}^+$ over $y_{u,i}^-$ given both the prompt and the persona. This can be instantiated as either offline DPO, where preference pairs are pre-collected, or online DPO, where new candidates are generated dynamically and preferences are updated on the fly. Both variants ensure that optimization is explicitly grounded in user personas, thereby complementing user-guided rewrites (Section 2.3) by extending alignment to the broader set of initial user requests when follow-up feedback is unavailable.

## 3 Experimental setup

### 3.1 Training Data Generation

**User Evaluation and Instruction-Following Tasks.** We build on the WildChat dataset, using 80% for training and reserving the rest for evaluation. To ensure quality, we exclude Midjourney-related instructions and retain only users with sufficient conversation history and meaningful feedback (details in Appendix C.1). To avoid training on GPT outputs as we use Llama for training, we construct a derived dataset, *WildLlamaChat*, which preserves only user messages. Assistant responses are

reconstructed by prompting Llama-3.1-8B-Instruct with the surrounding context, with details provided in Appendix C.3. For RLHI methods: (1) *RLHI with User-Guided Rewrites* uses Llama-3.1-8B-Instruct to generate user-based rewrites under sampling parameters $T = 0.6$ and $top\text{-}p = 0.9$. (2) *RLHI with User-Based Rewards* samples $N = 64$ responses per prompt from a curated pool of high-quality prompts using the same model and parameters, with the Athene-RM-8B reward model (Frick et al., 2024) providing user-based rewards.

**Reasoning Tasks.** Since no open-source dataset captures real human interactions in complex reasoning scenarios, we synthesize conversations by simulating users who ask math questions and point out model errors. These are based on the PRM800K dataset (Lightman et al., 2023), which includes MATH problems (Hendrycks et al., 2021), model-generated solutions, and step-level human correctness annotations. We randomly sample 10,000 erroneous solutions. In each conversation, the first turn presents a math problem, and the model replies with the dataset solution. In the second turn, the user makes comments such as "Step 3 seems incomplete or has an error" (details in Appendix C.4). Importantly, the simulated users only indicate where mistakes occur, without offering correct answers or detailed corrections, mimicking realistic user behavior. At training time, we apply *RLHI with User-Guided Rewrites* to revise unsatisfactory model outputs based on this feedback. Since the conversations are not tied to specific users, we do not incorporate user personas in this case.

## 3.2 TRAINING DETAILS

We initialize all models from Llama-3.1-8B-Instruct (Grattafiori et al., 2024). For RLHI methods: (1) *RLHI with User-Guided Rewrites* applies persona-conditioned DPO training, where we adopt a batch size of 64 and sweep over learning rates of $5 \times 10^{-7}$ and $1 \times 10^{-6}$. (2) *RLHI with User-Based Rewards* uses persona-conditioned online DPO training with batch size 32, learning rate $1 \times 10^{-6}$, and KL penalty $\beta = 0.01$. For instruction-following tasks, we perform early stopping using the same validation set as in Yu et al. (2025).

## 3.3 MODELS AND BASELINES

We compare RLHI against the following baselines: (1) **RL with Rewrites from Scratch**, which mirrors the *RLHI with User-Guided Rewrites* pipeline, but the model regenerates its responses without access to prior outputs or user feedback; (2) **RL with User-Agnostic Rewards**, which performs online DPO training on the same prompts used in *RLHI with User-Based Rewards*, but uses generic rewards that do not consider user personas; (3) **SFT with User-Guided Rewrites** and **SFT with User-Based Rewards**, which apply supervised finetuning on the chosen responses from our generated preference pairs; and (4) **RLHI w/o Quality Filtering**, which performs *RLHI with User-Guided Rewrites* but omits quality filtering of the rewrites.

## 3.4 EVALUATION SETTING

**User-Based Evaluation.** We introduce WILDCHAT USEREVAL, an LLM-based automated evaluation of personalization and instruction-following on real-world queries. We sample 100 users from the WildChat dataset with at least 10 conversations and substantial feedback. For each user, all but the last five conversations form the reference history, and the final five multi-turn dialogues are held out for evaluation. At each user turn in the held-out set, the evaluated model generates a response, which an OpenAI o3-based judge compares against the original ChatGPT response along three axes: (1) *Personalization*, where the judge first summarizes the user's persona from the reference history and decides which response better aligns with it; (2) *Instruction-Following*, assessing which response more faithfully follows the user's request and provides higher-quality content; and (3) *UserEval*, a holistic judgment simulating how a user would rate the responses, incorporating both aspects (1) and (2). See Appendix G for evaluation prompts. Model outputs are generated using decoding parameters $T = 0.6$ and $top\text{-}p = 0.9$ (consistent across evaluations below).

We consider two inference settings: (1) *Context-Only Inference*, where the model answers using only the ongoing multi-turn context, and (2) *Persona-Guided Inference*, where the evaluated model derives a persona from the reference history, and this persona is prepended to the user prompt, testing whether the model can both infer and leverage an explicit persona during generation.

Table 2: **User-Based Evaluations.** Win rates (%) judged by o3 against original ChatGPT responses on WILDCHAT USEREVAL. RLHI methods achieve substantial gains in personalization, instruction-following, and overall user preference compared to the seed model and baselines.

|  | Personalization | Instr-Following | UserEval |
|---|---|---|---|
| *Baselines* | | | |
| Llama-3.1-8B-Instruct | 38.2 | 30.6 | 32.5 |
| *+ Persona-Guided Inference* | 39.8 | 29.2 | 31.3 |
| RL with Rewrites from Scratch | 52.5 | 41.3 | 46.3 |
| *+ Persona-Guided Inference* | 54.6 | 40.4 | 47.3 |
| RL with User-Agnostic Rewards | 52.7 | 43.3 | 47.9 |
| *+ Persona-Guided Inference* | 54.2 | 42.8 | 48.4 |
| *RLHI* | | | |
| User-Guided Rewrites | 54.6 | 45.5 | 52.0 |
| *+ Persona-Guided Inference* | **62.5** | 44.5 | **54.9** |
| User-Based Rewards | 61.0 | **46.8** | 51.3 |
| *+ Persona-Guided Inference* | 62.3 | 44.7 | 52.5 |

Table 3: **Standard Evaluations.** Win rates (%) judged by GPT-4 Turbo on AlpacaEval2 and Arena-Hard. RLHI methods deliver large improvements over the seed model and baselines. *User-Based Rewards* beats or matches *RL with User-Agnostic Rewards* in this user-free setting.

|  | AlpacaEval2 | | Arena-Hard |
|---|---|---|---|
| *Standard models* | LC Win | Win | Score |
| Llama-3.1-8B-Instruct | 20.9 | 21.8 | 21.3 |
| RL with Rewrites from Scratch | 34.7 | 31.0 | 50.0 |
| RL with User-Agnostic Rewards | 77.0 | 73.3 | **64.4** |
| RLHI with User-Guided Rewrites | 35.2 | 38.5 | 51.2 |
| RLHI with User-Based Rewards | **77.9** | **83.4** | 64.3 |

To verify the reliability of LLM-based judgments, we also conduct a human study. We recruit $N = 10$ participants, each evaluating 50 randomly sampled turns under the same *UserEval* setting, with anonymized model identities and randomized response orders. Details are provided in Appendix E.

**Standard Evaluation.** We evaluate models on AlpacaEval 2.0 (Li et al., 2023; Dubois et al., 2024) and Arena-Hard (Li et al., 2024a), which are robust instruction following benchmarks that have a high correlation with human preferences. Evaluations are conducted with GPT-4 Turbo as the judge. AlpacaEval 2.0 includes both raw and length-controlled (LC) win rates.

**Reasoning Benchmarks.** We evaluate on OlympiadBench (He et al., 2024), Minerva (Lewkowycz et al., 2022), GPQA (Rein et al., 2024), and MMLU-Pro (Wang et al., 2024a), covering diverse reasoning challenges. For each problem, we sample $N = 50$ solutions and report average accuracy.

## 4 RESULTS

**User-Based Evaluation.** Table 2 provides results on WILDCHAT USEREVAL. RLHI methods consistently deliver strong improvements and outperform the baselines: *RLHI with User-Guided Rewrites* achieves the largest gains in personalization (+24.3) and overall improvement (+22.4), while *RLHI with User-Based Rewards* yields the strongest increase in instruction-following (+14.1). *RL with User-Agnostic Rewards* also significantly improves instruction-following but falls far behind RLHI in personalization (-8.3). Persona-guided inference enhances personalization, though sometimes at the cost of instruction-following. In the human study, *RLHI with User-Guided Rewrites* and *RLHI with User-Based Rewards* achieve win rates of 72.6% and 74.0% over Llama-3.1-8B-Instruct, confirming their effectiveness under direct human judgment.

**Standard Evaluation.** As shown in Table 3, RLHI achieves strong results in the standard user-free setting as well. *RLHI with User-Guided Rewrites* delivers large gains over Llama-3.1-8B-Instruct and outperforms *RL with Rewrites from Scratch*, although it lags behind online methods using reward models. This gap is likely due to the difference between training on multi-turn, real-user queries from WildChat and the single-turn, challenging prompts emphasized in these benchmarks.

However, *RLHI with User-Based Rewards* achieves 77.9% length-controlled win rate on AlpacaEval 2.0, outperforming *RL with User-Agnostic Rewards* and ranking above all RLHF methods on the leaderboard, and matches *RL with User-Agnostic Rewards* on ArenaHard in this user-free setting.

**Reasoning Benchmarks.** As shown in Table 3, *RLHI with User-Guided Rewrites* raises average accuracy from 26.5 to 31.8 across the four reasoning benchmarks. Among them, Minerva and OlympiadBench test math reasoning, while GPQA and MMLU-Pro evaluate advanced scientific and general-domain reasoning. Although training involves only math conversations, the gains transfer beyond math to broader reasoning tasks, indicating strong generalization. Notably, unlike methods that rely on verifiable rewards or detailed annotations, our setup involves simulated users who only flag mistakes without providing correct answers or fixes. Even such lightweight, realistic feedback improves reasoning, highlighting the effectiveness of learning from natural human interaction.

Table 4: **Performance on Reasoning Benchmarks.** *RLHI with User-Guided Rewrites* consistently improves over Llama-3.1-8B-Instruct across all tasks, yielding a +5.3 average gain.

|  | **Minerva** | **Olympiad** | **GPQA** | **MMLU-Pro** | **Avg.** |
|---|---|---|---|---|---|
| Llama-3.1-8B-Instruct | 20.2 | 14.5 | 26.3 | 44.9 | 26.5 |
| RLHI with User-Guided Rewrites | **25.4** | **18.4** | **33.1** | **50.1** | **31.8** |

## 4.1 Understanding Human Interaction and RLHI

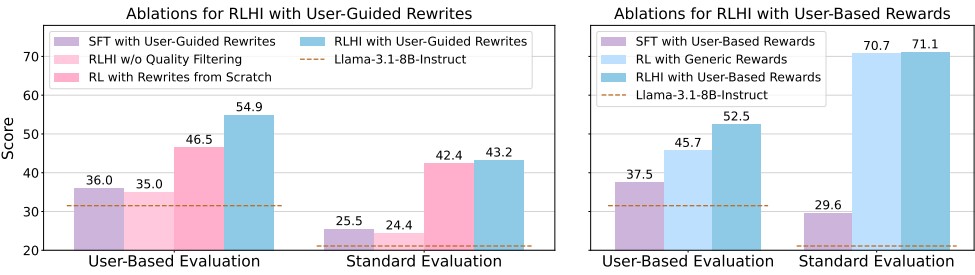

Figure 3: **Ablation Results.** User-Based Evaluation reports win rates on WILDCHAT USEREVAL, while Standard Evaluation averages AlpacaEval2 LC win rates and Arena-Hard scores. Both *RLHI with User-Guided Rewrites* and *RLHI with User-Based rewards* consistently outperform baselines.

**User-guided rewrites outperform regenerations by leveraging contextual feedback.** We compare *RLHI with User-Guided Rewrites* against *RL with Rewrites from Scratch*. When unsatisfactory responses are revised with user guidance rather than regenerated from scratch, the model benefits from direct, context-sensitive feedback that preserves the user's original intent while correcting specific deficiencies. This leads to stronger performance, as shown by (i) head-to-head rewrite comparisons, where User-Guided Rewrites achieves a 60.4% win rate under Athene-RM-8B, and (ii) training outcomes shown in Tables 2, 3, and Figure 3, where models trained with User-Guided Rewrites outperform repeated sampling on both user-based and standard evaluations, with notably larger gains in personalization (+7.9 points).

**User-based rewards capture long-term preferences for stronger alignment.** In *RLHI with User-Based Rewards*, the reward model ranks and selects responses conditioned on a persona derived from each user's long-term interaction history. By modeling such long-term preferences, user-based rewards guide the policy toward personalized behaviors that generalize across diverse queries. Compared to user-agnostic rewards, as shown in Tables 2, 3, and Figure 3, they substantially enhance personalization (+8.3 points), improve instruction-following and overall performance on real-world queries, and maintain competitive performance on standard benchmarks.

**RL outperforms supervised finetuning in learning from human interaction.** Figure 3 shows that SFT underperforms RL across both variants of our method and both evaluations. This gap arises because SFT relies only on positive examples and lacks gradient signals to distinguish good from bad responses. In contrast, RL methods such as DPO optimize policies over preference signals by leveraging both preferred and dispreferred examples, offering richer supervision regarding relative quality and more effectively aligning models with nuanced human preferences.

**Human interaction data is noisy and needs quality filtering.** The main challenge in RLHI is the noisiness of interaction data, which often includes low-quality prompts, harmful feedback, feedback inconsistent with earlier requests, or signals misaligned with common expectations. As shown in Figure 3, without filtering high-quality signals using reward models, *RLHI with User-Guided Rewrites* achieves only marginal gains of +2.5 and +3.3 points on user-based and standard evaluations. In contrast, filtering with reward models produces substantial improvements of +23.4 and +17.7 points, underscoring the critical role of quality control in leveraging human interaction for alignment.

**RLHI benefits from user diversity.** *RLHI with User-Guided Rewrites* learns from user conversations spanning 1268 users, each contributing only a few interactions. To isolate the role of diversity, we construct equally sized datasets but drawn from just 10 users with many conversations each. As shown in Figure 4, broader user diversity consistently improves win rates and scales more effectively, as the model learns to adapt to a wider range of preferences and interaction styles.

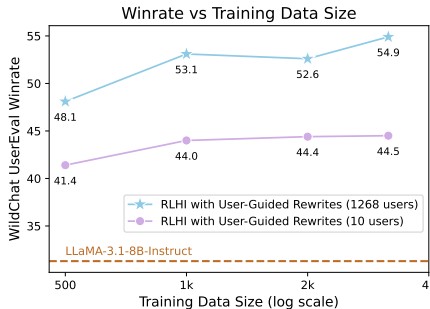

Figure 4: **Effect of user diversity on RLHI.** Training with 1268 diverse users outperforms training with 10 users of similar data size on WILDCHAT USEREVAL.

## 5 RELATED WORK

**Learning from Human Feedback.** Reinforcement Learning from Human Feedback (RLHF) trains a reward model on preference data and optimizes the base model with RL (Ziegler et al., 2019; Stiennon et al., 2020; Ouyang et al., 2022). Later work replaces explicit RL with direct preference optimization and related objectives for greater stability and efficiency (Rafailov et al., 2023; Ethayarajh et al., 2024; Azar et al., 2024). Beyond curated datasets, feedback is increasingly mined from post-deployment interactions: using user message classifiers (Hancock et al., 2019; Chen et al., 2024b; Don-Yehiya et al., 2024; Han et al., 2025), heuristics such as response length (Pang et al., 2023), or organic user signals like thumbs up/down and free-form comments (Jaques et al., 2020; Xu et al., 2023) to assess user attitude or satisfaction. These signals are then optimized via fine-tuning (Don-Yehiya et al., 2024) or other methods (Xu et al., 2023; Pang et al., 2023). Unlike prior work that relies on annotated labels or proxy signals, WildFeedback (Shi et al., 2024) and our RLHI approach learn directly from organic interactions. Our work goes further by modeling long-term user history, proposing user-based rewards, demonstrating stronger performance on standard benchmarks, and enabling user-based evaluation.

**Personalizing Language Models.** Personalization aims to adapt LMs to user preferences through retrieval, prompting, representation learning, or RLHF (Zhang et al., 2024). Retrieval and prompting approaches incorporate user information as external memory (Mysore et al., 2023; Salemi et al., 2024) or as persona/profile context (Jiang et al., 2023). Representation-learning methods encode traits in model parameters (Tan et al., 2024) or embeddings (Chen et al., 2025). RLHF-style personalization uses user information as reward signals to align LLMs with personalized preferences: works explore conditioning on multiple reward dimensions (Jang et al., 2023; Yang et al., 2024; Li et al., 2024b; Shenfeld et al., 2025), decoupling generation dynamics from user utility (Chen et al., 2024a), generalized system messages during training (Lee et al., 2024), or aligning models through a user-specific latent variable model (Poddar et al., 2024). Our RLHI framework explicitly connects long-term personas with turn-level preferences and optimizes on organic interactions, yielding stronger personalization and better instruction-following.

## 6 CONCLUSION

In this paper, we make the case for the improvement of models by learning from real-world human interaction. We present a concrete method, Reinforcement Learning from Human Interaction (RLHI), a simple and scalable framework for learning directly from in-the-wild user conversations utilizing long-term conversation history and organic natural-language feedback. RLHI provides

clear improvements when measured at the *user* level compared to strong baselines, where utilizing organic feedback is shown to improve both non-reasoning and reasoning tasks. Looking forward, we see opportunities to extend RLHI with human-in-the-loop learning, richer and safer reward modeling, privacy-preserving personalization, and broader modality and task coverage. Importantly, we believe using RLHI within an online learning loop, where a continually updating deployed model learns from its organic interactions, would bring major gains compared to the fixed training data setup in our experiments. We hope these findings encourage a shift toward learning from real-world human interaction to build capable, personalized assistants that improve over time.

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

# A  EXAMPLES OF USER CONVERSATION DATA

As described in Section 2.2, each multi-turn conversation begins with an *initial request*, which accounts for 27.07% of all user messages. User follow-ups are then classified into four types: *new requests* (40.40%), *re-attempts with feedback* (26.51%), *re-attempts without feedback* (4.77%), and *positive feedback* (1.25%). Below, we provide real examples from WildChat for each type.

## A.1  NEW REQUESTS

We present three examples of *new requests*: (1) In the first, the *initial request* is only a greeting or non-question, followed by a substantive query. (2) In the second, the user makes a different translation request. Although it is still thematically related to the initial one (both about Viennese translation), the specific content changes, so it is classified as a *new request*. (3) In the third, the user shifts topics entirely, making a request unrelated to the original.

---

**NEW REQUEST – EXAMPLE 1**

**User message:** hi

**Assistant message:** Hello! How can I assist you today?

**User message:** write a long poem about AI

---

**NEW REQUEST – EXAMPLE 2**

**User message:** Hello! Please tell me how to say "I am also a kid" in Viennese Dialect

**Assistant message:** In Viennese dialect, you would say "I bin aa a Kind."

**User message:** So, how is your day? In Viennese

---

**NEW REQUEST – EXAMPLE 3**

**User message:** how begin conversation with girl that i like

**Assistant message:** As an AI language model, I do not have personal experiences or emotions, but here are some tips on how to start a conversation with someone you like:
1. Start with... (713 characters skipped here)

**User message:** teach me speak english

---

## A.2  RE-ATTEMPTS WITH FEEDBACK

We present three examples of *re-attempts with feedback*: (1) In the first, the user repeats the request for more ideas but adds the instruction "make them creative," offering explicit guidance for improvement. (2) In the second, the user reframes the request by introducing a new dimension, asking how social classes perceive and interpret an act, which meaningfully extends the original query. This reveals the perspective the user values and signals what could be incorporated directly into future responses. (3) In the third, the user corrects the assistant by pointing out a length requirement ("2000+ words"), providing clear feedback on where the response fell short and how to adjust. Interestingly, the initial request specified "a 2000 word essay," without mentioning "2000+" or "about 2000." This inconsistency may reflect a linguistic habit, but it also helps the assistant better interpret the user's intended word limit in future interactions.

---

**RE-ATTEMPT WITH FEEDBACK – EXAMPLE 1**

**User message:** give me 20 ideas for themes for a summer camp for children aging from 4 to 18

**Assistant message:** 1. Nature and Outdoor Adventure... (642 characters skipped here)

**User message:** give me 20 more make them creative

---

**RE-ATTEMPT WITH FEEDBACK – EXAMPLE 2**

**User message:** What cultural or traditional significance does wearing a hood hold in this particular world?

**Assistant message:** In this particular world, wearing a hood holds cultural and traditional significance in several ways:
1. Protection from the elements... (1946 characters skipped here)

**User message:** How do different social classes in this world perceive and interpret the act of wearing a hood?

---

**RE-ATTEMPT WITH FEEDBACK – EXAMPLE 3**

**User message:** write a 2000 word essay on how political ideologies influence modern political thinking

**Assistant message:** Introduction:
Political ideologies have been around for centuries, and they have influenced the way people think about politics and governance... (9406 characters skipped here)

**User message:** i said 2000+ word

---

A.3  RE-ATTEMPTS WITHOUT FEEDBACK

We present two examples of *re-attempts without feedback*. (1) In the first, the user repeats the same request but merely adds "Best," signaling dissatisfaction without offering actionable guidance. (2) In the second, the user initially asks for 20 more examples with the instruction "make them creative," but later repeats the request for 20 more without providing any new signals, which is then classified as a *re-attempt without feedback*.

---

**RE-ATTEMPT WITHOUT FEEDBACK – EXAMPLE 1**

**User message:** give me some promt for ask Ai to design to me wallpeper phone

**Assistant message:** Sure, here are some creative prompts you can use to ask an AI to design a wallpaper for your phone:... (1698 characters skipped here)

**User message:** give me some Best promt for ask Ai to design to me wallpeper phone

---

**RE-ATTEMPT WITHOUT FEEDBACK – EXAMPLE 2**

**User message:** give me 20 ideas for themes for a summer camp for children aging from 4 to 18

**Assistant message:** 1. Nature and Outdoor Adventure... (642 characters skipped here)

**User message:** give me 20 more make them creative

**Assistant message:** 1. Fantasy Island Adventure... (702 characters skipped here)

---

> **User message:** give me 20 more

## A.4 Positive feedback

Positive feedback is rare, accounting for only 1.25% of user messages, since users often end the conversation once satisfied. Below, we show one example where the user expresses appreciation with a simple "thank you."

---

**POSITIVE FEEDBACK – EXAMPLE 1**

**User message:** What is the difference between Hinduism and Buddhism ?

**Assistant message:** As an AI language model, I don't believe in beliefs and religions. However, here are the major differences between Hinduism and Buddhism:
1. Belief in God:... (1267 characters skipped here)

**User message:** Thank you

**Assistant message:** You're welcome!

---

## B Details of User Conversation Data Analysis

We visualize the prompt embeddings calculated in Section 2.2. This demonstrates that real-world human interaction data are more diverse than existing preference datasets.

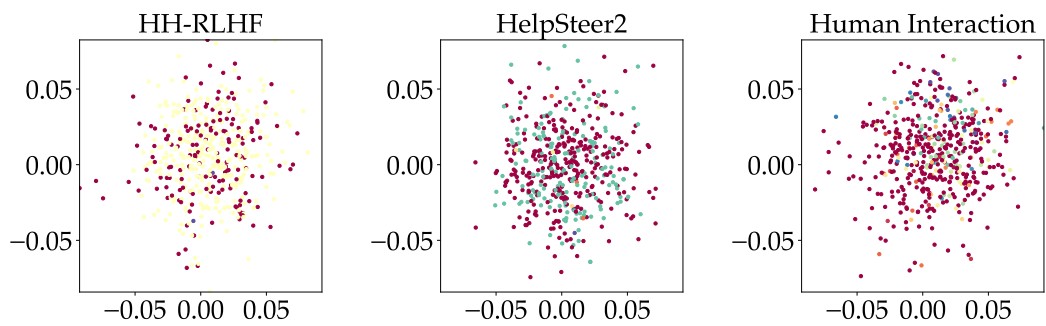

Figure 5: Visualization of context embeddings across preference datasets: the two annotated human feedback datasets, HH-RLHF and HelpSteer2, and our human interaction dataset used for RLHI.

## C Details of User Conversation Data Processing

### C.1 Details of Training Conversation Filtering

To ensure data quality and relevance, we apply several filtering steps to the WildChat-1M dataset (Zhao et al., 2024b), before using RLHI to learn from the user conversations:

1. Exclude non-English prompts using the provided language annotations.
2. Remove Midjourney-related instructions, which typically begin with: "As a prompt generator for a generative AI called 'Midjourney', you will create image prompts ...".
3. Retain only users with at least three conversations, ensuring enough context to infer a persona.
4. Discard users with more than 100 conversations, as they are often associated with program-generated instructions that are low quality and misaligned with real human needs.

5. Exclude conversations with more than 10 turns to maintain task focus and coherence.

6. Use an LLM to filter for users who provide meaningful feedback.

## C.2 DETAILS OF PREFERENCE PAIR FILTERING

To improve the quality of preference pairs used for optimization, we adopt RIP's filtering techniques (Yu et al., 2025) with the following thresholds:

1. **Rejected response length** $\geq 1878$: Following Yu et al. (2025), we treat rejected response length as a proxy for prompt quality. Low-quality prompts (unclear, ambiguous, or conflicting) tend to produce short, uninformative responses, which correlate with weaker performance (Zhao et al., 2024a; Yuan et al., 2024).

2. **Rejected response reward** $\geq -1$: We use Athene-RM-8B (Frick et al., 2024) to assign user-based rewards, ensuring rejected responses still meet a minimal quality threshold.

3. **Reward gap** $\leq 1$: Large reward gaps often arise from low-quality prompts that allow multiple interpretations. By restricting the gap between chosen and rejected responses, we favor prompts that elicit consistent, high-quality outputs.

## C.3 DETAILS OF CONSTRUCTING WILDLLAMACHAT

To avoid training on GPT outputs as we use Llama for training, we construct a derived dataset, *WildLlamaChat*, which preserves only user messages. Assistant responses are reconstructed by prompting Llama-3.1-8B-Instruct with the surrounding context. We don't just use the previous context—we also provide the subsequent user messages. This is crucial because the follow-up feedback naturally constrains the reconstruction to be consistent with that feedback. For example, if a user's next message says "rewrite it, do not consider it from the angle of A," the reconstructed response will be generated to consider angle A, making it appropriate for the user's feedback. We empirically found that reconstruction with only previous context does produce misaligned responses, but including future context significantly improves alignment. Additionally, we apply reward model-based filtering to ensure quality and relevance. While we acknowledge this approach has limitations, it's necessary to avoid training on GPT outputs while still leveraging valuable user feedback from WildChat.

We use Llama-3.1-8B-Instruct to both generate the training data (reconstructions and rewrites) and serve as the model being trained. This is essentially a self-improvement setup where the model learns from its own responses and user feedback on those responses. If we were to train a different model, we would use that specific model to generate its own training data, maintaining consistency between the data generation and training processes.

## C.4 DETAILS OF SYNTHESIZING MATH CONVERSATIONS

Since no open-source dataset captures real human interactions in complex reasoning scenarios, we synthesize conversations by simulating users who ask math questions and point out model errors. These are based on the PRM800K dataset (Lightman et al., 2023), which includes MATH problems (Hendrycks et al., 2021), model-generated solutions, and step-level human correctness annotations. From this corpus, we randomly sample 10,000 erroneous solutions and the corresponding questions.

Each synthetic conversation begins with a math problem ending with the instruction: "Please reason step by step, and put your final answer within \boxed{}." The model then replies with the dataset solution, consisting of multiple steps annotated with human judgments of correctness. In the next turn, the user identifies the first incorrect step and provides natural-language comments such as "Step 3 seems incomplete or has an error." If the final answer is correct despite earlier mistakes, the user adds a qualifier such as "... though your final answer is correct." In this way, the simulated users only indicate where mistakes occur, without offering correct answers or detailed corrections, mimicking realistic user behavior.

## D  ADDITIONAL RESULTS ON WILDCHAT USEREVAL

In Table 5, we show results on WILDCHAT USEREVAL, breaking down the overall win rates from Table 2 into performance on initial turns and following turns. This decomposition reveals how well models handle first attempts compared to user follow-ups later in the conversation. RLHI methods continue to outperform baselines across both settings, with the strongest gains from User-Guided Rewrites, which achieves 60.3% on initial turns and 52.6% on follow-up turns when combined with Persona-Guided Inference, leading to the best overall UserEval score of 54.9%. These results highlight that RLHI consistently enhances model responses throughout multi-turn interactions.

Table 5: **User-Based Evaluations with Turn-Level Breakdown.**  Win rates (%) judged by o3 against original ChatGPT responses on WILDCHAT USEREVAL.  This table expands upon the UserEval results in Table 2 by separately reporting performance on initial user turns ("Initial") and follow-up turns ("Follow-up"), providing a more detailed view of how models handle different types of requests.

|  | UserEval (Initial) | UserEval (Follow-up) | UserEval |
|---|---|---|---|
| Llama-3.1-8B-Instruct | 36.3 | 30.9 | 32.5 |
| *+ Persona-Guided Inference* | 33.0 | 30.6 | 31.3 |
| RL with Rewrites from Scratch | 47.2 | 45.9 | 46.3 |
| *+ Persona-Guided Inference* | 46.7 | 47.6 | 47.3 |
| RL with User-Agnostic Rewards | 50.6 | 46.8 | 47.9 |
| *+ Persona-Guided Inference* | 50.6 | 47.5 | 48.4 |
| RLHI with User-Guided Rewrites | 57.0 | 49.9 | 52.0 |
| *+ Persona-Guided Inference* | **60.3** | **52.6** | **54.9** |
| RLHI with User-Based Rewards | 50.3 | 51.7 | 51.3 |
| *+ Persona-Guided Inference* | 54.7 | 51.6 | 52.5 |

## E  DETAILS OF THE HUMAN STUDY

We recruited 10 participants to evaluate the model outputs, none of whom were paper authors. Participants gave informed consent under an IRB-exempt protocol and were compensated at standard rates. Each participant evaluated 50 items sampled from held-out WildChat conversations. For each item, we showed the multi-turn context up to the user's last message and a short persona summary derived from that user's past conversations (the same prompt used in our automated "UserEval" setup). Raters then saw two anonymized responses (A/B) from different models in random order and selected which they would prefer as the user, considering both how well it followed instructions and how personalized it was to the user's history. We compared Llama-3.1-8B-Instruct to each RLHI variant. In total, the study produced 500 pairwise judgments: 250 for Llama-3.1-8B-Instruct vs. RLHI with User-Guided Rewrites and 250 for Llama-3.1-8B-Instruct vs. RLHI with User-Based Rewards.

The results of our human study strongly supported the findings from our automated evaluation. Human evaluators preferred the responses from our RLHI with User-Guided Rewrites method 72.6% of the time over the baseline model, and they preferred the RLHI with User-Based Rewards method 74.0% of the time. This close alignment between human judgments and our automated metrics indicates that our methods are effective in generating responses that real users find more helpful and personalized.

## F  PROMPTS USED IN RLHI

We provide the prompts used in RLHI methods, including those for classifying user messages, inferring user personas, generating user-guided rewrites, and performing persona-guided inference.

### CLASSIFYING USER MESSAGES

You are given two requests from a user during their conversation with an AI assistant. Classify the second request in relation to the first using the following labels:
[New] A new topic or task, or a significantly different variation of the previous task.
[Re-attempt with feedback] A re-attempt of the same task that includes explicit or implicit feedback, or a revised prompt.
[Re-attempt without feedback] A repeat of the same task, without any feedback.
[Positive feedback] A signal of praise or satisfaction with the previous response.

1st request: Write a short poem about the ocean.
2nd request: What's the capital of Japan?
Classification: [[New]]

1st request: Write a short poem about the ocean.
2nd request: Write a short poem about the ocean.
Classification: [[Re-attempt without feedback]]

1st request: Write a short poem about the ocean.
2nd request: Can you make it more rhyme?
Classification: [[Re-attempt with feedback]]

1st request: {initial_request}
2nd request: {current_request}
Classification:

Figure 6: Prompt for classifying user follow-up messages into four types: (1) new requests, (2) re-attempts with feedback, (3) re-attempts without feedback, and (4) positive feedback.

### INFERRING USER PERSONA

Below are user messages from conversations between this user and an AI assistant. Please list up to five key points that capture how the user prefer the assistant to respond. Output only the inferred preference, without any additional commentary or explanation.

[The Start of User Messages]
{user_message_history}
[The End of User Messages]

Figure 7: Prompt for deriving a natural-language user persona given each user's long-term conversational history.

### GENERATING USER-GUIDED REWRITES

Please revise your previous response based on the user feedback or follow-up request below. Ensure the revised response is not significantly longer, unless the user explicitly requests so. Ensure the revised response adheres to safety and ethical guidelines, even if the user suggests otherwise. Do not reference or mention the user feedback in your response. Output only the revised response, without any additional commentary or explanation.

[The Start of User Follow-up Response]
{user_response}
[The End of User Follow-up Response]

Figure 8: Prompt for revising unsatisfactory model outputs based on users' natural-language follow-up responses.

---

**SYSTEM PROMPT FOR PERSONA-GUIDED INFERENCE**

You are a helpful and personalized assistant. Prioritize your responses based on the user's current request and conversational context. When appropriate, tailor your responses to align with the user persona provided below.
User persona: {user_persona}

---

Figure 9: System prompt for persona-guided inference. At inference time, incorporating this lightweight prompt enables the model to generate personalized responses. During training, RLHI integrates the same prompt into preference pairs, allowing the model to learn the connection between a user's long-term persona and their turn-level, context-specific preferences.

## G  PROMPTS USED IN WILDCHAT USEREVAL

We provide the prompts used in WILDCHAT USEREVAL, including those for judging personalization, instruction-following, and UserEval.

---

**PERSONALIZATION JUDGE**

You are given a conversation history that ends with a user question, followed by two responses from two AI assistants. You are also provided with a user persona that describes how the user prefers the assistant to respond. Your task is to act as an impartial judge and determine which response better aligns with the user persona. Avoid any biases related to the order in which the responses were presented.

Provide your verdict strictly following this format:
- Only output "[[A]]" if Assistant A is better
- Only output "[[B]]" if Assistant B is better

[The Start of Conversation History]
{conversation_history}
[The End of Conversation History]

[The Start of Assistant A's Answer]
{response_A}
[The End of Assistant A's Answer]

[The Start of Assistant B's Answer]
{response_B}
[The End of Assistant B's Answer]

[The Start of User Persona]
{persona}
[The End of User Persona]

---

Figure 10: Prompt for the personalization judge in WILDCHAT USEREVAL. The judge first summarizes the user's persona from the reference history using the prompt in Figure 7, and then applies this prompt to determine which response aligns better with it.

---

**INSTRUCTION-FOLLOWING JUDGE**

You are given a conversation history that ends with a user question, followed by two responses from two AI assistants. Your task is to act as an impartial judge and determine which response better follows the user's instructions and provides a higher-quality answer. Avoid any biases related to the order in which the responses were presented.

---

Provide your verdict strictly following this format:
- Only output "[[A]]" if Assistant A is better
- Only output "[[B]]" if Assistant B is better

[The Start of Conversation History]
{conversation_history}
[The End of Conversation History]

[The Start of Assistant A's Answer]
{response_A}
[The End of Assistant A's Answer]

[The Start of Assistant B's Answer]
{response_B}
[The End of Assistant B's Answer]

Figure 11: Prompt for the instruction-following judge in WILDCHAT USEREVAL, determining which response better follows the user's instructions and provides a higher-quality answer.

**USEREVAL JUDGE**

You are given a conversation history that ends with a user question, followed by two responses from two AI assistants. You are also provided with a user persona that describes how the user prefers the assistant to respond. Your task is to act as an impartial judge, simulating how the user would evaluate the responses. Specifically, determine which response better follows the user's instructions, provides a higher-quality answer, and aligns with the user persona. Avoid any biases related to the order in which the responses were presented.

Provide your verdict strictly following this format:
- Only output "[[A]]" if Assistant A is better
- Only output "[[B]]" if Assistant B is better

[The Start of Conversation History]
{conversation_history}
[The End of Conversation History]

[The Start of Assistant A's Answer]
{response_A}
[The End of Assistant A's Answer]

[The Start of Assistant B's Answer]
{response_B}
[The End of Assistant B's Answer]

[The Start of User Persona]
{persona}
[The End of User Persona]

Figure 12: Prompt for the UserEval judge in WILDCHAT USEREVAL. The judge first summarizes the user's persona from the reference history using the prompt in Figure 7, and then applies this prompt to determine which response better follows the user's instructions, provides a higher-quality answer, and aligns with the user's persona.

## H   THE USE OF LARGE LANGUAGE MODELS

In accordance with the ICLR 2026 Author Guide, we disclose our use of Large Language Models (LLMs): after completing the draft of the paper, LLMs were used to polish the writing. They were not used for any other purpose.

