# OpenReview forum: "The Era of Real-World Human Interaction: RL from User Conversations"
_ICLR.cc/2026/Conference — Submitted to ICLR 2026_

### Official Review · Reviewer_w7qf · 2025-10-17

**Soundness:** 3
**Presentation:** 2
**Contribution:** 3
**Rating:** 4
**Confidence:** 2

**Summary:**

The paper presents RLHI (Reinforcement Learning from Human Interaction), a framework for post-training personalization of language models through continuous learning from real-world user interactions. The authors introduce two methods for this process: User-Guided Rewrites, which revises unsatisfactory model outputs based on users' follow-up responses, and User-Based Rewards, which ranks responses based on a reward model conditioned on long-term user personas. Both methods aim to align model outputs with users' ongoing, evolving preferences. Evaluations show that these methods outperform baseline models on personalization and instruction-following tasks on datasets like AlpacaEval 2.0 and Arena-Hard, with additional improvements in reasoning tasks. The results highlight the effectiveness of learning directly from organic human interaction in achieving personalized and contextually aligned models.

**Strengths:**

RLHI is an advancement in aligning language models with personalized user preferences by directly learning from ongoing user conversations. And the research question is important. By focusing on real-world human interaction, the paper pushes the field towards a more scalable, personalized approach to model alignment.

**Weaknesses:**

1. Complexity in Application: While the methods work well in controlled settings, they involve managing user persona conditioning, which could become cumbersome for models with large vocabularies or diverse users. Is it feasible to apply these methods at scale without significant overhead, especially when it comes to managing persona conditioning across millions of users?
2. Method Simplicity: The methods proposed, while effective, are relatively straightforward. The first method uses persona rewriting and the second applies persona-conditioned reward modeling. These are not particularly novel compared to existing techniques, such as RLHF. The simplicity of the methods raises questions about why such straightforward approaches result in such impressive outcomes (77.9 on AlpacaEval2.0 and 64.3 on ArenaHard).

**Questions:**

1. In Table 3, it seems that RL with User-Agnostic Rewards setting achieves a relatively good performance without a human persona. How to explain it?
2. In Table 3, RL with Rewrites from Scratch and User-Agnostic Rewards seems to have a higher LC win compared to Win on AlpacaEval2. Does it indicate that these two methods may favor shorter responses?

---

> ### Author Response · Authors · 2025-11-22
> **Author Response**
>
> Thank you for your insightful review and for recognizing that RLHI pushes the field towards a more scalable, personalized approach to model alignment. We are happy to address your questions and suggestions below:
>
> ---
>
> ## W1. Scalability Concerns for Managing User Personas
>
> We appreciate the concern regarding scalability. Our approach is designed to be highly efficient by using **simple, natural-language summaries** for user personas, rather than complex user models or latent embeddings as described in the related works section. These text-based personas are straightforward to manage and can be updated periodically, such as after every 100 user conversations, which avoids significant computational overhead. This makes our method practical and scalable for millions of users while still delivering strong personalization results.
>
> ---
>
> ## W2. Strong Results Despite Method Simplicity
>
> Thank you for the insightful comments. We agree that some components of our RLHI framework build on established principles in the field. However, our core contribution is proposing a new learning paradigm and showing its surprising effectiveness. We ask a fundamental question: can we significantly improve models by learning directly from noisy, in-the-wild user conversations? Our work is the first to provide an end-to-end framework to answer this question, serving as a strong proof-of-concept for learning from human interaction.
>
> The impressive results are not due to complex methods, but from applying our approach to the **rich and previously under-utilized data** of real user conversations. As our analysis shows, organic user interactions are more diverse and grounded than static datasets. Our work suggests the path to better models may not require more complex algorithms, but rather a shift toward learning from more natural and scalable sources of supervision. Our extensive ablation studies support this, highlighting the **key factors for success**: the need for quality filtering, the advantage of RL over supervised finetuning, the effectiveness of organic feedback and long-term personas, and the benefits of user diversity. By showing that even a straightforward approach can achieve state-of-the-art results, we hope to encourage the community to further explore this promising research direction.
>
> ---
>
> ## Q1. Strong Performance Without User Personas
>
> Thank you for the insightful question. It is true that RL with User-Agnostic Rewards (RLHF + our data generation in the WildChat setting) serves as a very strong baseline, achieving competitive performance on standard (user-free) benchmarks. This is expected, as these benchmarks are designed to assess general instruction-following capabilities rather than personalized user alignment.
>
> Our primary contribution, however, is to improve model performance on user-centric metrics that reflect **real-world user** experiences. When evaluated in this context using WildChat UserEval, the user-agnostic approach falls significantly behind our RLHI methods in personalization, with RLHI using User-Based Rewards outperforming it by 8.1 points. This shows that while standard RLHF is effective for general capabilities, it does not capture the long-term, nuanced preferences of individual users, which is a central goal of our work. The fact that our method performs well (matching RLHF) on user-free evaluations demonstrates that we can achieve strong personalization without compromising core instruction-following abilities. The fact that our method performs well (matching RLHF) on general benchmarks shows that we can deliver strong personalization and user-level improvements **without sacrificing instruction-following capabilities** in single-turn, non-personalized scenarios.
>
> ---
>
> ## Q2. Methods May Favor Shorter Responses
>
> Yes, your observation is correct. The higher LC Win rate for RL with Rewrites from Scratch and RL with User-Agnostic Rewards on the AlpacaEval2 benchmark suggests that these models tend to produce shorter responses than the reference responses used for evaluation in the benchmark. In contrast, the RLHI methods tend to produce slightly longer responses.

---

### Official Review · Reviewer_6TvU · 2025-11-01

**Soundness:** 2
**Presentation:** 3
**Contribution:** 3
**Rating:** 6
**Confidence:** 4

**Summary:**

This paper introduces RLHI (Reinforcement Learning from Human Interaction), a framework that learns directly from user interaction in real conversations. RLHI includes User-Guided Rewrites and User-Based Rewards, leveraging both long-term preference and turn-level feedback to achieve continual personalized alignment. Experimental results demonstrate that RLHI not only improves performance in user-based evaluations but also achieves notable gains in instruction-following and reasoning tasks.

**Strengths:**

1. **Grounding in Real-World Human Preference**: RLHI learn human preference directly from natural user interactions, which provide more diverse signals than than existing curated datasets.

2. **Comprehensive Experiments and Strong Performance**: RLHI methods are evaluated across user-based evaluation, instruction-following, and reasoning tasks, achieving significantly performance improvements. Further ablation studies validate the effectiveness and contribution of each proposed component.

3. **Insightful Analysis** : This paper provides meaningful anaysis and insight about the properties of current human interaction data, demonstrating that feedback often occur in user conversations and highlighting these statics are likely to change with the evolve of LLMs.

**Weaknesses:**

1. The authors use Llama-3.1-8B to reconstruct responses while retaining the original user feedback (Line 269). However, since different models may behave differently, the preserved feedback may no longer be accurate or applicable, potentially misguiding the model.

2. The term “user-based rewards” can be misleading, as it suggests a reward model that scores based on user persona. In practice, the framework just samples multiple responses conditioned on the user persona and selects among them using a reward model.

3. The experiments on reasoning tasks assumes that human feedback is correct (synthesize conversations based on PRM dataset). However, human users may not be experts and could introduce errors, leading the model to learn incorrect preferences. In my view, RLHI is not well suited for objective tasks like math reasoning.

4. The proposed method heavily relies on external reward models for data filtering and for providing high-quality supervision. Its improvements on non-personalized tasks are limited (comparing RLHI with user-based and user-agnostic rewards on user-free evaluation), suggesting that much of the gain may stem from the reward model rather than from the RLHI framework.

5. I have some concerns that the paper presents the method as reinforcement learning, while the experiments are conducted solely using DPO.

**Questions:**

See above section

---

> ### Author Response · Authors · 2025-11-22
> **Author Response (Part I)**
>
> Thank you for your insightful review and for recognizing RLHI’s novelty, comprehensive experiments, strong performance, and insightful analysis. We are happy to address your questions and suggestions below:
>
> ---
>
> ## W1. Potential Mismatch Between Reconstructed Responses and Original User Feedback
>
> Thank you for raising this concern. We would like to clarify two important aspects of our reconstruction process, which were included in our updated Appendix C.3 (highlighted in blue).
>
> 1. When we reconstruct assistant responses using Llama-3.1-8B-Instruct, we don't just use the previous context—we also provide the subsequent user messages. This is crucial because the follow-up feedback naturally **constrains the reconstruction to be consistent with that feedback**. For example, if a user's next message says "rewrite it, do not consider it from the angle of A," the reconstructed response will be generated to consider angle A, making it appropriate for the user's feedback. We empirically found that reconstruction with only previous context does produce misaligned responses, but including future context significantly improves alignment. Additionally, we apply reward model-based filtering to ensure quality and relevance. While we acknowledge this approach has limitations, it's necessary to avoid training on GPT outputs while still leveraging valuable user feedback from WildChat.
>
> 2. We use Llama-3.1-8B to both generate the training data (reconstructions and rewrites) and serve as the model being trained. This is essentially a **self-improvement setup** where the model learns from its own responses and user feedback on those responses. If we were to train a different model, we would use that specific model to generate its own training data, maintaining consistency between the data generation and training processes.
>
> ---
>
> ## W2. Terminology Around “User-Based Rewards”
>
> This concern is mostly due to misunderstandings. As stated in Section 2.4, our reward model explicitly **conditions on the user persona** when scoring responses. Specifically, the reward model evaluates each candidate as $r(y_{u,i}^{(n)} \mid x_{u,i}, p_u)$, where $p_u$ is the user persona (Equation 2).
>
> This is further emphasized in the text: "we generate preference pairs by first sampling $N$ candidate responses, then evaluating them with a reward model that explicitly conditions on the corresponding user persona" and "if long-term interactions indicate that a user favors answers with numbers, statistics, and concrete evidence, the reward model will assign higher scores to responses that not only meet general quality criteria but also reflect these user-specific characteristics".
>
> Thus, both the candidate sampling process and the reward scoring are persona-conditioned, making "user-based rewards" an accurate description. The ablation comparing against RL with User-Agnostic Rewards further validates that persona-conditioned reward modeling is the key distinction and drives the performance gains.
>
> ---
>
> ## W3. Assumption of Correct Human Feedback in Reasoning Tasks
>
> Thank you for raising this. We agree that our reasoning experiments rely on simulated feedback and that real users can be wrong. Since our primary goal is personalization and instruction-following based on real conversations, the reasoning component is not the main focus of the paper.
>
> We used PRM-based data to mimic a common real-world pattern—users only indicate which step contains an error without offering corrections, reflecting **realistic behavior** where users identify issues but may not know the correct solution. We acknowledge this setup has *limitations*: the feedback isn't from real users, is mostly correct, lacks diversity in phrasing, and contains no persona signals. We will make these limitations clear in the updated paper.
>
> To reduce harm from bad feedback, our pipeline already uses **quality filtering**. In future work, we plan to collect real reasoning chats, pair RLHI with self-checks, and weight feedback by confidence or agreement. This should better address objective domains where correctness matters more than preference.

---

> > ### Author Response · Authors · 2025-11-22
> > **Author Response (Part II)**
> >
> > ## W4. Reliance on Reward Models
> >
> > While RLHI uses a reward model for quality control and supervision, its main contribution is showing how to effectively use **signals from real-world user interactions** to improve personalization and instruction-following. The key is not just the reward model, but using it to learn from real-world conversations and **conditioning it on users’ long-term history** to connect long-term and turn-level preferences.
> >
> > Our user-based evaluations in Table 2 show this clearly: when the reward model incorporates the user’s persona (RLHI with User-Based Rewards), the personalization score increases substantially (+8.1 points) compared to when it does not (RL with User-Agnostic Rewards). This directly isolates the benefit of our proposed method beyond the reward model. Moreover, in RLHI with User-Guided Rewrites, the learning signal is derived from using the user’s own natural follow-up messages to refine responses.
> >
> > RLHI is the first RL pipeline to demonstrate the surprising effectiveness of learning directly from human interaction. We acknowledge that on non-personalized tasks, RLHI matches or slightly exceeds RLHF’s performance. However, our main focus is user-based evaluation and **surpassing RLHF in personalization and instruction-following on real-world user queries**. The fact that our method performs well on general benchmarks shows that we can deliver strong personalization and user-level improvements **without sacrificing instruction-following capabilities** in single-turn, non-personalized scenarios.
> >
> > ---
> >
> > ## W5. RL Framing and DPO Experimental Setup
> >
> > Our use of DPO is a direct implementation of our reinforcement learning framework. DPO is a widely recognized RL method for LLM alignment that optimizes the model's policy using preference data as an implicit reward signal. Many contemporary papers in this domain build their RL framework solely on DPO. Our paper uses this technique to effectively learn from the noisy, real-world user interactions that we argue are essential for the next generation of model improvement. It is also important to note that our approach is modular and would function with other RL methods that learn from preference comparisons.

---

### Official Review · Reviewer_YARd · 2025-11-02

**Soundness:** 2
**Presentation:** 3
**Contribution:** 2
**Rating:** 4
**Confidence:** 3

**Summary:**

# Summary
A post-training framework is introduced to learn directly from in-the-wild user–LLM conversations instead of pre-annotated expert feedback. Two complementary variants are presented: **RLHI with User-Guided Rewrites** (convert user follow-ups that revise unsatisfactory answers into preference pairs) and **RLHI with User-Based Rewards** (rank candidates using a reward model conditioned on a summarized user “persona” from long-term histories). On **user-based evaluations** built from WildChat conversations, both RLHI variants improve personalization and instruction-following over strong baselines; **RLHI-User-Based Rewards** attains **77.9% length-controlled win rate on AlpacaEval 2.0** and strong Arena-Hard results under **GPT-4-Turbo** judging (Table 3). **RLHI-User-Guided Rewrites** also raises accuracy on four reasoning benchmarks from **26.5 to 31.8** despite training only on math conversations (Table 4). Ablations indicate benefits from user guidance, interaction diversity, RL over SFT, and quality filtering.

**Strengths:**

# Strengths
* **Realistic supervision source:** Moves beyond curated annotator pipelines to leverage in-situ signals (SAT/DSAT, rewrites, follow-ups).

* **Compelling empirical improvements (user-based & standard).**

    • On **WILDCHAT USEREVAL**, RLHI variants improve personalization and overall user preference (Table 2).

    • On **AlpacaEval 2.0** and **Arena-Hard**, RLHI-User-Based Rewards achieves 77.9% LC win and 83.4 Arena-Hard win (Table 3).

    • On reasoning, **RLHI-User-Guided Rewrites** lifts average accuracy **26.5 to 31.8** across Minerva, OlympiadBench, GPQA, and   MMLU-Pro (Table 4).

**Weaknesses:**

# Weaknesses

1. **Evaluator and judge bias**

   * *Example (benchmarks):* MT-Bench uses **GPT-4o** as judge; AlpacaEval 2 and Arena-Hard use **GPT-4-Turbo**, all closed-source LLM judges.
   * *Example (user eval):* RLHI reports user-based results judged by **OpenAI o3** on WILDCHAT USEREVAL, again relying on a closed-source judge.

2. **Persona inference via LLM**

   * *Example:* RLHI **prompts an LLM to summarize each user’s persona** from long-term histories to steer both training (persona-conditioned DPO) and inference—useful but introduces risk of hallucination/leakage without audits.

**Questions:**

## What Could Be Improved.
1. **Add strong baselines**
* **Teacher-distill baseline**: SFT on *ChatGPT/strong model* responses for the same turns. This tests whether mined preferences beat a simple, distillation.
* **Ground-truth tasks**: Include **GSM8K**, **MATH**, **MBPP/HumanEval** with exact-match verify gains on **verifiable** reasoning/coding.

2. **Robust judging**
* Evaluate with **multiple independent judges** (closed and open), randomize templates, and report **win-rate consistency** across judges. Include **anti-style** prompts (brevity/conciseness constraints).

**Details Of Ethics Concerns:**

## Responsible research practice.
The data originates from real user interactions (WildChat) rather than synthetic or crowdsourced text.

---

> ### Author Response · Authors · 2025-11-22
> **Author Response**
>
> Thank you for your insightful review and for recognizing that our paper addresses an important challenge and presents strong experiments. We are happy to address your questions and suggestions below:
>
> ---
>
> ## W1. Evaluator and Judge Bias
>
> Our evaluation approach is **aligned with the requirements of standard benchmarks** such as AlpacaEval 2.0 and Arena-Hard, which use LLM judges because tasks involving instruction-following lack single correct answers, and large-scale human evaluation is impractical. In addition to these standard methods, we introduced WildChat UserEval, which is better suited to our focus on real-world user experiences. Most importantly, we conducted a **human study** to validate our findings, which confirmed that our automated judge's conclusions are consistent with real human preferences, directly addressing concerns about potential bias.
>
> ---
>
> ## W2. Persona Inference via LLM
>
> To address the concern regarding hallucination, we treat persona inference strictly as a summarization task rooted in the user’s existing conversation history. We prompt the model to extract only explicit preferences and styles present in the context window, minimizing the chance of the model inventing false details. Additionally, our training pipeline includes quality filtering; if an inaccurate persona leads to a poor response, that data point is identified and excluded from the training set.
>
> Regarding data leakage, the inferred personas are used as temporary context for the current session rather than being memorized into the model’s permanent weights. We can apply filtering to prevent the inclusion of sensitive personal details, and the persona is designed to capture simple **stylistic preferences**, such as a preference for concise or detailed answers, to improve the user’s experience rather than to create a comprehensive or private profile. This ensures the model learns the general ability to adapt without retaining specific user data.
>
> ---
>
> ## Q1. Add Teacher-Distill Baselines & Ground-Truth Tasks
>
> Thank you for this suggestion. While distilling from a stronger model like ChatGPT is a valuable baseline, our work aims to test a different hypothesis. We want to show that learning directly from a user's natural, ongoing feedback provides a unique signal for improvement. This method captures a user's specific and evolving preferences from their conversation history, a dynamic process that simply copying a stronger model's static, one-size-fits-all answers would miss. Our goal is to demonstrate the value of the interaction signal itself for adapting and improving a model’s personalization and instruction-following.
>
> We fully agree that evaluating on tasks with verifiable, ground-truth answers is essential. This is why we included a suite of rigorous reasoning benchmarks in our study. Our experiments measure performance on OlympiadBench, Minerva, GPQA, and MMLU-Pro, all of which are **challenging, exact-match evaluations** for mathematics and science. The results in Table 3 show that our method significantly boosts accuracy on these tasks, indicating that the improvements reflect genuine gains in reasoning ability.
>
> ---
>
> ## Q2. Robust Judging
>
> In standard benchmarks like AlpacaEval 2.0 and Arena-Hard, our approach aligns with common practices that use GPT-4 Turbo judges. In our WildChat UserEval, we designed the evaluation to directly address the robustness concern by using **separate judges** for different criteria. We include an instruction-following judge that is prompted to ignore style and focus solely on whether the model follows the user's direct commands, including constraints like requests for brevity. This ensures we are measuring true instruction-following rather than a learned conversational style. This is complemented by our personalization judge, which evaluates alignment with the user's broader preferences. Our positive results on both of these distinct evaluations show that the model is not simply overfitting to a single style, but is effectively handling both personalization and explicit instructions.

---

### Official Review · Reviewer_Fryz · 2025-11-03

**Soundness:** 2
**Presentation:** 3
**Contribution:** 2
**Rating:** 4
**Confidence:** 3

**Summary:**

The paper presents RLHI, a post-training paradigm that learns directly from real-world user interactions to align large language models with evolving, long-horizon preferences. RLHI converts organic conversations into preference signals via two complementary routes: (1) User-Guided Rewrites, which transforms user follow-ups into improved revisions to form preference pairs; and (2) User-Based Rewards, which ranks multiple candidates using a reward model conditioned on a natural-language user persona. Over user-based evaluation, standard instruction-following benchmarks, and reasoning suites, RLHI delivers consistent gains over baselines. Ablations highlight the importance of leveraging explicit feedback, RL over SFT on interaction data, and the effect of quality filtering.

**Strengths:**

1. The authors clearly pinpoint an important gap that current alignment underutilizes rich multi-turn user feedback. They leverage long-horizon personas and turn-level corrections to drive personalized learning.

2. The paper uses two complementary, lightweight routes (user-guided rewrites and user-based rewards) under a DPO-style objective, making the approach easy to integrate and broadly usable even with sparse explicit ratings.

3. It demonstrates effectiveness on both user-based evaluations and standard public leaderboards, with consistent gains and informative ablations across diverse tasks and domains.

**Weaknesses:**

1. Limited novelty. The user-guided rewrites track prior WildChat-style pipelines [1,2]; the user-based rewards overlap with user-conditioned reward models [3,4] and conditioning LM/RM/judges on user summaries [5,6,7]. If those lines cannot solve the paper’s target problem, the manuscript should specify why; otherwise, stronger baseline comparisons are needed.

2. Human study is under-specified. Results are summarized in one sentence with no details.

3. Reasoning part uses PRM data to simulate conversations without analyzing its limitations. Also, if personas are not used at inference, the setup collapses to vanilla DPO, weakening the core claim.

[1] WildFeedback: Aligning LLMs With In-situ User Interactions And Feedback. arXiv:2408.15549.

[2] User Feedback in Human-LLM Dialogues: A Lens to Understand Users But Noisy as a Learning Signal. arXiv:2507.23158.

[3] Personalized Language Modeling from Personalized Human Feedback. arXiv:2402.05133.

[4] Personalizing Reinforcement Learning from Human Feedback with Variational Preference Learning. arXiv:2408.10075.

[5] RLPF: Reinforcement Learning from Prediction Feedback for User Summarization with LLMs. arXiv:2409.04421.

[6] Learning to summarize user information for personalized reinforcement learning from human feedback. arXiv:2507.13579.

[7] Can LLM be a Personalized Judge? arXiv:2406.11657.

**Questions:**

1. What's the core novelty comparing with the papers listed in the weaknesses?

2. Please provide more details about the human study and the experiments on reasoning benchmarks. (See Weaknesses)

3. How does the system adapt to truly new users? Do the identified preference dimensions and feedback types generalize under cold-start conditions?

---

> ### Author Response · Authors · 2025-11-22
> **Author Response (Part I)**
>
> Thank you for your insightful review and for recognizing that our paper addresses an important challenge and presents strong experiments. We are happy to address your questions and suggestions below:
>
> ---
>
> ## W1 & Q1: Unclear Novelty Comparing With Prior Papers
>
> We updated the Related Works and Conclusion sections (highlighted in blue) to expand discussions and highlight our novelty.
>
> **Core novelty:** This paper aims to answer two key questions that prior work has not addressed: Can we learn directly from real-world user conversations? And can this meaningfully improve personalization, instruction-following, and reasoning? To answer these questions, we (1) introduce a unified, end-to-end RLHI framework that learns directly from user conversations and connects long-term persona-level preferences with turn-level signals. (2) RLHI is the first RL pipeline demonstrating the surprising effectiveness of learning from human interaction, with ablations that isolate what matters (quality filtering, RL vs. SFT, and user diversity).
>
> **Core differences:** Our paper reports significantly stronger results compared with prior WildChat-style pipelines [1,2] (which show no or limited improvement), due to methodological innovations such as persona-conditioned preference optimization. Our method learns from real-world user conversations, whereas [3–7] learn from simulated users in very different settings with very different reward formulations.
>
> **How RLHI differs from WildChat-style feedback pipelines [1,2]:**
>
> - [1,2] have no user consistency in their optimization because they do not use **user-level information**. They do not improve personalization and do not match our goal of user-level gains. In contrast, we address this by obtaining user personas and performing persona-conditioned preference optimization, effectively linking long-term user personas to turn-level preferences. This connects persistent preferences to immediate corrections during training and inference, not just during data construction.
>
> - While [2] emphasizes that implicit feedback is noisy and inconsistently helpful, our pipeline addresses this directly with reward-model and RIP-style filtering, demonstrating that in-the-wild feedback can be an effective learning signal.
>
> - RLHI achieves **significantly stronger performance on standard evaluations**: we obtain 77.9 on AlpacaEval2 and 64.3 on Arena-Hard (34.7 and 50.6 for user-guided rewrites), while [1] reports 34.2 and 32.9, respectively. We show clear improvements, whereas [2] reports minimal gains and concludes that WildChat user feedback is “noisy as a learning signal.”
>
> - We establish a **user-based evaluation**, assessing personalization and instruction-following on real-world user queries. This is our main experimental focus and is absent in [1,2]. We show that user-guided rewrites achieve even stronger user-based improvements by learning from user follow-ups, with RLHI outperforming RLHF at the user level.
>
> **How RLHI differs from user-conditioned RM/LM approaches [3–7]:**
>
> - Supervision in all these papers relies on curated preference data rather than **organic multi-turn interaction**.
>
> - [3,4] condition the reward on user models or latent user variables. In contrast, RLHI achieves strong results by simply conditioning on natural-language user personas, using personas for three roles simultaneously: (i) guiding data generation/selection from organic conversations, (ii) conditioning preference optimization, linking the same persona representation across the pipeline, and (iii) guiding personalized generation at inference time. This triple use provides a direct mechanism for **long-horizon preferences to shape turn-level learning signals**.
>
> - [5,6] optimize the quality of user persona summaries. In contrast, RLHI shows that personas distilled from long histories measurably improve instruction-following and personalization when embedded directly into the preference objective and generation, rather than being used only as side information for a reward or judge.
>
> - While the “personalized judge” work [7] examines evaluation reliability, we pair an LLM judge with a human study and report results on both user-based evaluations and popular standard benchmarks.
>
> ### References:
> [1] WildFeedback: Aligning LLMs With In-situ User Interactions And Feedback. arXiv:2408.15549.
>
> [2] User Feedback in Human-LLM Dialogues: A Lens to Understand Users But Noisy as a Learning Signal. arXiv:2507.23158.
>
> [3] Personalized Language Modeling from Personalized Human Feedback. arXiv:2402.05133.
>
> [4] Personalizing Reinforcement Learning from Human Feedback with Variational Preference Learning. arXiv:2408.10075.
>
> [5] RLPF: Reinforcement Learning from Prediction Feedback for User Summarization with LLMs. arXiv:2409.04421.
>
> [6] Learning to summarize user information for personalized reinforcement learning from human feedback. arXiv:2507.13579.
>
> [7] Can LLM be a Personalized Judge? arXiv:2406.11657.

---

> ### Author Response · Authors · 2025-11-22
> **Author Response (Part II)**
>
> ## W2 & Q2: Insufficient Details on Human Study
>
> Thank you for the feedback. We added an Appendix E (highlighted in blue) to provide more details.
>
> We recruited 10 participants to evaluate the model outputs, none of whom were paper authors. Participants gave informed consent under an IRB-exempt protocol and were compensated at standard rates. Each participant evaluated 50 items sampled from held-out WildChat conversations. For each item, we showed the multi-turn context up to the user’s last message and a short persona summary derived from that user’s past conversations (the same prompt used in our automated “UserEval” setup). Raters then saw two anonymized responses (A/B) from different models in random order and selected which they would prefer as the user, considering both how well it followed instructions and how personalized it was to the user’s history. We compared Llama-3.1-8B-Instruct to each RLHI variant. In total, the study produced 500 pairwise judgments: 250 for Llama-3.1-8B-Instruct vs. RLHI with User-Guided Rewrites and 250 for Llama-3.1-8B-Instruct vs. RLHI with User-Based Rewards.
>
> The results of our human study strongly supported the findings from our automated evaluation. Human evaluators preferred the responses from our RLHI with User-Guided Rewrites method 72.6% of the time over the baseline model, and they preferred the RLHI with User-Based Rewards method 74.0% of the time. This close alignment between human judgments and our automated metrics indicates that our methods are effective in generating responses that real users find more helpful and personalized.
>
> ---
>
> ## W3 & Q2: Unanalyzed Limitations and Insufficient Details in PRM-Simulated Reasoning Experiments
>
> We want to clarify that the reasoning experiments are not our primary focus. Our main contribution is demonstrating RLHI for personalization and instruction-following using real user conversations. The reasoning experiments are included as supplementary evidence that RLHI can also improve performance on reasoning benchmarks when provided with knowledgeable user feedback.
>
> We use PRM data to simulate conversations, as there is **no open dataset** capturing real human interactions in complex reasoning scenarios. The synthetic conversations we generate are intentionally simple—users only indicate which step contains an error without offering corrections, reflecting realistic behavior where users identify issues but may not know the correct solution. We acknowledge this setup has *limitations*: the feedback isn't from real users, the wording is less diverse, and there is no persona signal. We will make these limitations clear in the updated paper.
>
> Even without personas at inference time, the experiment does not reduce to “vanilla DPO”: the preference pairs are derived from rewrites based on error mentions within a conversational context, rather than from static annotations or direct comparisons.

---

> > ### Author Response · Authors · 2025-11-22
> > **Author Response (Part III)**
> >
> > ## Q3: Adaptation to Truly New Users and Generalization Under Cold-start Conditions
> >
> > **How the system adapts to truly new users:** In our user-based experiments, we work with “truly new users.” The users in the WildChat UserEval set are entirely separate from those in the training set, and therefore represent completely new users to the models.
> >
> > **How preference dimensions and feedback types generalize under cold-start conditions:** To evaluate cold-start conditions, we conduct experiments with zero user history. This setup corresponds exactly to the “Context-Only Inference” setting in WildChat, which does not use Persona-Guided Inference or any personas derived from reference histories. As shown in the table below, under cold-start conditions, our method significantly outperforms the baselines.
> >
> >
> > |                          | Personalization | Instr-Following | UserEval |
> > |--------------------------|----------------:|-----------------:|---------:|
> > | *Baselines*              |                 |                  |          |
> > | Llama-3.1-8B-Instruct    | 38.2            | 30.6             | 32.5     |
> > | RL with Rewrites from Scratch | 52.5      | 41.3             | 46.3     |
> > | RL with User-Agnostic Rewards | 52.7      | 43.3             | 47.9     |
> > | *RLHI*                   |                 |                  |          |
> > | User-Guided Rewrites     | 54.6            | 45.5             | **52.0** |
> > | User-Based Rewards       | **61.0**        | **46.8**         | 51.3     |
> >
> >
> > **Explanations**: Our system performs well even with new users because it is trained on conversations spanning a broad and diverse population. This enables the model to learn general patterns of user preferences and feedback that generalize across many users. Under cold-start conditions, the system can still respond effectively because the conversational context often implicitly reveals user preferences, even without any persona derived from prior history. RLHI can then gradually update the persona during inference, allowing the model to adapt over time while collecting requests and feedback to support self-evolving personalization. As a result, the model has a strong starting point for any new interaction and quickly adapts to a new user’s style and needs as the conversation progresses.

---

### Meta-Review · Area_Chair_difg · 2026-01-01

**Summary:**

While the paper proposes a promising Reinforcement Learning from Human Interaction (RLHI) framework utilizing WildChat data to enhance personalization, reviewers expressed common concerns that the reliance on static, offline conversation logs fails to sufficiently validate the claims of improvement in dynamic, real-world human-model interaction.

**Reviewer Concerns:**

The rebuttal successfully clarified the technical implementation of the persona-conditioned reward model and the filtering of noisy WildChat data (addressing initial methodological questions), but the reviewers' fundamental concern regarding the absence of a live, human-in-the-loop user study to verify online adaptation remains outstanding.

**Reviewer Scores:**

Reviewers may likely maintain their score or very slightly raise score, but a core question remains w.r.t. paper lacking human study live evaluation to back claims or method robustness.

---

### Decision · Program_Chairs · 2026-01-26

Reject